# Factors Influencing Spiritual Health among Nursing Students in the Prolonged COVID-19 Situation

**DOI:** 10.3390/ijerph20043716

**Published:** 2023-02-20

**Authors:** Juhyun Jin

**Affiliations:** Institute of Nursing Science, College of Nursing, Daegu Catholic University, Daegu 42472, Republic of Korea; dominicajin@cu.ac.kr

**Keywords:** COVID-19, pandemic, spiritual health, nursing student, South Korea

## Abstract

The COVID-19 pandemic is not only an epidemiological crisis but also a spiritual health crisis that affects nursing students. Spiritual health is essential in maintaining and promoting physical and mental health to achieve happiness, potential, meaning, and purpose of life even during a pandemic. This descriptive cross-sectional study aimed to examine factors affecting spiritual health of nursing college students. The study adheres to the Strengthening the Reporting of Observational studies in Epidemiology (STROBE) guidelines. A total of 219 nursing students from three nursing colleges in Metropolitan D city participated in the study through an online Google Form questionnaire from 2–18 September 2021. The mean score of spiritual health was 96.98 ± 11.54 (out of 120 points); spiritual health was significantly positively correlated with life satisfaction and academic performance (*p* < 0.001) and negatively correlated with academic stress (*p* < 0.001). Factors significantly affecting spiritual health were academic stress (ß = −2.21, *p* = 0.045), life satisfaction (ß = 3.85, *p* < 0.001), and academic performance; below score of 3.0 (ß = −2.08, *p* = 0.039). The explanatory power of these effects was 30.7%. As a future professional nurse who will work in the clinical field where the demand for the spiritual care of patients is increasing, it is necessary to develop and apply a curriculum that can improve the spiritual health of nursing students.

## 1. Introduction

Humans have been navigating a different world since the outbreak of COVID-19 in December 2019 [1]. The COVID-19 pandemic, which has been ongoing for three years at the time of writing, is not only an epidemiological crisis but also a health crisis that creates diverse sociopsychological problems, such as stress, anxiety, depression, trauma, panic, insomnia, and boredom [2,3], and the young population is particularly at greater risk of being affected by these problems [3,4,5,6]. The young population can experience various social problems, financial difficulties due to limited economic activities, stress and learning gaps due to online education, reduced face-to-face relationships, routinization of contact-free relationships, and discrimination and loathing against confirmed cases of COVID-19 [4,5,6,7]. This eventually hinders them from living a healthy life in terms of physical, mental, and social state [3,4,5,6]. In fact, many previous studies have reported a health crisis in the younger generation considering the pandemic situation. According to a 2022 survey conducted by the Korean government, one in five adults are at risk of developing depression and in their 20s, including university students; rates of depression have doubled, and suicidal ideation has increased more than 2.5-fold compared with before the pandemic [3]. In the case of nursing students, even before the pandemic, the academic burden was already higher than that of other major students due to credit management, practice, and theoretical classes linked to the Korea Nursing Licensing Examination. However, the stress created by this workload, combined with the uncertainty of future employment further increased with the COVID-19 pandemic [1,8,9,10]. Previous studies on nursing students in the U.S. [8], China [4], Taiwan [5], Turkey [6], and Israel [7] have reported fear of infection, powerlessness, depression, academic stress, decreased academic achievement, changes in mental health, and reduced adjustment to university and life satisfaction. Moreover, high levels of stress and decreased life satisfaction are expected to have a negative impact on the overall health of nursing students [11,12]. Health is not only a disease-free state but also a harmonious state of well-being in which physical, mental, and social health linked to each other are integrated by spiritual health [9] and nurses’ high health level is an essential resource to provide holistic care. Therefore, a comprehensive understanding of health and self-health management should be incorporated into future nursing education [5,8,13]. As one of the key aspects of health, spiritual health is important for nursing students’ wellbeing and their future career success. Spiritual health is also an essential factor in maintaining and promoting physical and mental health to achieve the happiness, potential, meaning, and purpose of life of nursing students [13]. Spiritual health is the latest dimension of health that coordinates the physical, psychological, and social aspects [9]. Although there is no universally accepted definition of spiritual health, researchers have offered some definitions. For instance, spiritual health was defined as a dynamic state that continues to grow, create, and live well with happiness from within in order to achieve the meaning and purpose of life by recognizing one’s life connected to the absolute including heaven and ancestors, establishing a harmonious relationship with others, and overcoming spiritual obstacles [12]. In this context, the spirituality of individuals and communities can support mental health, which includes the ability of healthy human relationships, overcoming negative emotions such as anxiety and depression, and growing and developing themselves [5,8,13]. In modern society, the number of people with chronic diseases increases due to the development of medical technology, and accordingly, there is an increased interest in spiritual health that regulates and integrates mental, social, and physical health, as well as a demand for spiritual care that increases spiritual health [12]. However, spiritual care tends to be recognized only as hospice care and the role of religious, whereas the need and understanding of a high level of nurses’ spiritual health, which is a priority for providing high-quality spiritual care for patients, is insufficient. Meanwhile, in Korea, the concept of spiritual health has not been clearly defined as it is combined with both spirituality and spiritual well-being [12]. In current nursing studies, spiritual health has a multidimensional and comprehensive meaning that transcends specific religions, and the terms spirituality, spiritual well-being, and subjective sense of well-being are used interchangeably [10,12]. Notwithstanding, a few spiritual-health-related studies exist, namely on the relevance of spiritual health, depression, anxiety, clinical practice stress, and health-promoting behaviors among Taiwanese nursing students [14], and another on a spiritual health intervention for undergraduate nursing students using a spiritual learning program [10]. Spiritual health has also been perceived as being the core of physical, mental, and social health, bringing a positive change to health behavior, particularly in the field of nursing where self-care and holistic care for patients are essential [12]. Furthermore, the studies reported spirituality and spiritual well-being of nursing students have confirmed the relationship with positive affect, gratitude disposition, flow, and self-esteem [15]; self-resilience, self-efficacy, and interpersonal competence [16]; stress-related academic and clinical areas [14,17,18]; anxiety and depression [8]; and quality of life [17,19] before and after the COVID-19 pandemic. For nursing students, fear of infection, academic stress, academic performance, and life satisfaction are expected to be negatively impacted as theory and clinical practice combine in the COVID-19-altered educational system. Nursing is a discipline associated with human life, and nursing students experience high levels of academic stress [10,14,15,17]; the higher the levels of perceived stress, the lower the level of spirituality and spiritual health [10,11,17]. Nevertheless, high levels of academic performance form a positive factor in the spirituality of nursing students [11] and quality of life, including life satisfaction has also been identified as a positive factor influencing nursing students’ spiritual health [17,19]. Spiritual health is essential and useful in restoring nursing students’ physical, mental, and social health, enabling them to perform holistic care as specialized nurses in the future [11,12]. Therefore, owing to the ongoing COVID-19 pandemic, this study attempted to identify the level of the spiritual health of Daegu-based nursing students, a cohort who entered university in 2020 and experienced the pandemic. Psychological and emotional factors, such as the fear of COVID-19, and study-related factors, such as academic stress, academic achievement, and overall life satisfaction, were selected as the main variables, and their impact on spiritual health was investigated. The results of this study will be beneficial to nursing educators as a basis for identifying ways to promote the spiritual health of nursing students, and for developing the practical intervention program and its applications.

## 2. Materials and Methods

### 2.1. Study Design and Participants

This study employed a cross-sectional research design. The participants of this study were nursing students who understood the purpose of the research and voluntarily agreed to participate; students taking a temporary leave of absence were excluded. The G*Power statistical software (https://www.psychologie.hhu.de/arbeitsgruppen/allgemeine-psychologie-und-arbeitspsychologie/gpower) package was used to calculate the required sample size; for an effect size of 0.15 and 95% power, the software suggested that 217 participants were required. A total of 219 sets of data were collected and used for the final analysis, which is an appropriate number of samples for this study. 

### 2.2. Data Collection

A convenience sampling was applied to select nursing students from three consenting nursing colleges in Daegu Metropolitan City from 2 to 18 September 2021. The researcher first explained the purpose of the study and autonomy of participation to the student representative of each academic year through a phone call, email, or Social Networking Service and sent the research recruitment notice containing a Google Forms link, through an online messenger. The Google Forms link to the survey was posted in the group chat rooms of each class, and participants voluntarily accessed the web to read the study’s description and consent form, agreed to participate, and participated anonymously. The study description included the purpose, ethical considerations for participants (confidentiality, anonymity, informed written consent, and the right to withdraw), and the research process. The Google Forms survey was set to terminate automatically if the participant did not consent. The completion time of the survey was between 20 and 25 min.

### 2.3. Instruments

The questionnaires used in this study elicited information on fear of COVID-19, academic stress, academic performance, satisfaction of life, spiritual health, and sociodemographic information. All questionnaires were developed in Korean and were completed by Korean-speaking participants.

#### 2.3.1. The Korean Version of Fear of COVID-19 Scale (KF-COVID-19S)

Fear of COVID-19 was measured with the KF-COVID-19S, translated by Hwang et al. (2021) [20], and originated from the Fear of COVID-19 scale developed by Ahorsu et al. [2]. A total of seven items were present using a five-point Likert scale, and the score ranged from 7–35 points, with higher scores indicating a higher level of Fear of COVID-19. The internal consistency (Cronbach’s α) was 0.88 in Hwang et al.’s study [20] and 0.82 in this study. 

#### 2.3.2. Scale of Academic Stress (SAS) 

Academic stress was measured with SAS developed for adolescents by Park and Park (2012) [21], and included “parents,” “teacher,” and “oneself” as the causes of academic stress. However, in this study, based on previous studies regarding nursing students, 15 questions were measured using only “oneself” as a source of academic stress. The scale contained 15 items on a five-point Likert scale and the score ranged from 15 to 75 points, with higher scores indicating a higher level of academic stress. Cronbach’s α was 0.85 in Lee and Park’s study for nursing students [18], and 0.87 in this study.

#### 2.3.3. Academic Performance 

In this study, academic performance was evaluated as a subject score obtained by learning, and as a comprehensive grade point average (GPA) for the entire course of the previous semester from the time of data collection. The question was, “What was your GPA in the first semester of 2021?” Academic grading in South Korea depends on the type of school in which it is involved. In the University, the grade comprises a letter type, similar to American schools. The letter grades can add up to numbered averages ranging from 0.0 to 4.5 with higher scores indicating a higher level of academic performance [22]. 

#### 2.3.4. The Korean Version of Satisfaction with Life Scale (SWLS) 

Satisfaction of life was measured with the Korean version of the SWLS translated by Hong et al. [23], and originated from the SWLS developed by Pavot and Diener [24]. It included five items on a seven-point Likert scale and the score ranged from 5 to 35 points, with higher scores indicating a higher level of life satisfaction. Cronbach’s α was 0.84 in Hong et al.’s study and 0.84 in this study [23]. 

#### 2.3.5. Spiritual Health Scale-Short Form (SHS-SF) 

Participants’ religious affiliation in the demographic characteristics is not a measure of spirituality. Therefore, this study adapted SHS-SF developed by Hsiao et al. (2013) which had high validity and reliability for measuring the spiritual health of students and registered nurses in Taiwan which has a Buddhist and Confucian culture similar to that of Korea [25]. There are several instruments of spirituality in previous studies including the Spiritual Well Being Scale (SWB), the JAREL spiritual well-being scale, the Functional Assessment of Chronic Illness Therapy Spiritual Well-being scale (FACIT-Sp), and Spiritual Index of Well-being (SIWB). These instruments generally have common shortcomings. The majority of scale items are focused on religiosity, which may not be applicable to non-religiosity and all instruments were developed based on western Judeo-Christian society [25]. This study utilized the SHS-SF which can measure the concept of spiritual health distinct from spirituality and spiritual well-being; moreover, it was first translated into Korean and applied. The original version comprises 47 items (2005), whereas the short version consists of 24 simplified items and uses a five-point Likert scale. The score ranged from 24 to 120 points, with higher scores indicating a higher level of spiritual health. The *SHS-SF* was translated into Korean according to the methods and procedures established by the World Health Organization [26]. The researcher first translated the original English questionnaire into Korean. Following this, two nursing professors, experienced in spirituality-related research and fluent in English, reviewed, evaluated, and revised the Korean version of the questionnaire. The revised version was commissioned to a specialized translation site (Editage; https://www.editage.com/ (accessed on 1 July 2021) for a translation back into English. One American professor and two nursing professors proficient in English compared the back-translated English version to the original one. They reaffirmed that the translation was accurate without any change in meaning. Expert validity tests were then performed at the pre-testing stage. Five nursing professors with experience in studying the spirituality and psychology of nursing students confirmed the content validity. The latter was calculated using the Scale Level Content Validity Index (S-CVI) with a four-point scale from 1 = not at all relevant to 4 = very relevant. S-CVI scores of all items were higher than 0.75, satisfying content validity. Cronbach’s α was 0.92 in Hsiao et al.’s study [25] and 0.89 in the present study.

#### 2.3.6. General Characteristics 

To determine the participants’ general characteristics, their sociodemographic information such as sex, age, religion affiliation, socioeconomic status, monthly allowance, residential environment, grade, reason for entering nursing school, leave absence, club activities, clinical practice, type of lectures, number of consultations with a professor, major satisfaction, were collected. 

### 2.4. Statistical Methods

The collected data were analyzed using SPSS Windows 23.0 (SPSS, Chicago, IL, ISA). The analysis included descriptive statistics (i.e., percentages, means, and standard deviations), which estimated the general characteristics and features of main variables, independent sample t-tests, ANOVA, Pearson’s correlation coefficient, and hierarchical multiple regression. The significance level was set to a standard of α < 0.05. 

### 2.5. Ethical Considerations

The data collection process received approval from the Bioethics Committee of Daegu Catholic University (CUIRB-2021-0049-01). The participants received all information about this study online and checked a box for consent to participate. Data collection was conducted on weekends to avoid disturbing students’ lectures, and the researcher’s mobile number and e-mail were circulated so that any questions during participation could be addressed at any time.

## 3. Results

There were statistically significant differences in spiritual health in the 23 years of age and older group; that with the highest socioeconomic status; and that with the highest satisfaction with academic major. Religion was a statistically significant variable; however, there were no significant differences between the religious groups (Table 1).

Leave of absence and participation in student clubs were found to be significant variables. All the mean scores of the main variables were confirmed to be above the average score for each measurement instrument (Table 2). 

Spiritual health showed a moderate negative correlation with academic stress (r = 0.307), a strong correlation with life satisfaction (r = 0.43), and a weak correlation with academic achievement (r = 0.19) (Table 3). 

However, spiritual health was not correlated with the fear of COVID-19. As shown in Table 4, In step 1, participants’ age, religion, socioeconomic status, leave of absence, experience in club activities, and major satisfaction among demographic factors, which statistically significantly differed according to spiritual health were entered into Model 1 for hierarchical regression and nominal variables were dummy coded. The explanatory power of Model 1 was 22.6% and the model had a significant fit (F = 7.94, *p* < 0.001). In step 2, fear of COVID-19, academic stress, academic performance levels, and life satisfaction were additionally entered as independent variables after controlling for general characteristics in Model 2. The spiritual health of Protestant (β = 0.18) and Catholic (β = 0.15) students was higher than in the non-religious group (referenced value) and that of individuals with grades below 3.0 (β =−0.14) was lower than for those with grades over 4.0 (referenced value). Academic stress (β = −0.15) and performance (<3.0 grades) were negative predictors, and satisfaction of life (β = 0.27), and being religious (Protestant and Catholic) were positive predictors of spiritual health. The explanatory power of Model 2 was 30.7%, and the model had a significant fit (F = 6.83, *p* < 0.001).

## 4. Discussion

The mean score of the fear of COVID-19 among Korean nursing students was higher than that for nursing students in Turkey [6], Turkish adults aged 18–25 years [27], and Korean adults [20] when measured using the same instrument. 

This result can be attributed to the fact that 59 % of the participants in this study were third- and fourth-year students who participated in clinical practice and had a higher fear of contracting COVID-19 than those in the first and second year of university [1]. Evidently, the fear of infection was high among Korean nursing students participating in clinical practice in the hospitals where patients diagnosed with COVID-19 were being treated. The mean score of academic stress was lower than that for third- and fourth-year students, measured in Korea that are using the same instrument [28]. These scores differed as this study represented first to fourth-year students, whereas the previous study included only third- and fourth-year students with higher academic stress than that of their first- and second-year counterparts [29]. Increased academic stress associated with online classes in students after the pandemic was identified in previous research, and comparative and analytical studies on the academic stress of students experiencing in-person classes are needed in the future [1,17,28]. The mean score of life satisfaction was lower than the life satisfaction of nursing students before the outbreak of COVID-19 [30]. In the context of the prolonged pandemic, it is expected that overall life satisfaction has decreased due to fear of infection, stress, anxiety, and depression in social life and clinical practice [1,8,31,32]. However, life satisfaction was higher than that of general university students across a similar time period [31]. This result may be attributed to nursing students having relatively secure employment and career choices compared with other students, who, due to the economic downturn, have higher levels of stress regarding career and job readiness [32]. The mean score of spiritual health of nursing students in this study was lower than the scores reported in previous studies [14,33] and Chiang et al. [5] studies that measured the spiritual health of nursing students in Taiwan before the COVID-19 outbreak. The study of Chiang et al. was a longitudinal study from 2017 to 2019 and reports that the level of spiritual health decreases as the academic year of nursing students increases [5]. Therefore, a longitudinal study is needed to confirm the level of spiritual health that changes as the academic year increases. Additionally, it was reported that the level of spiritual health of new nurses was higher than that of fourth-year nursing students, suggesting that nursing students may need customized programs for an improvement in their spiritual health [5].

Regarding the general characteristics of nursing students, there were differences in their level of spiritual health according to age, socioeconomic status, religion, student club activities, and satisfaction with academic major. This was similar to previous studies where the same general characteristics displayed varying levels of spiritual health [11,33,34]. In particular, students with no religious affiliation had the lowest level of spiritual health. This means that not only religiosity but also non-religiosity is an important factor in spiritual health; therefore, education and training about the spiritual realm, beliefs, values, and ethics derived from human conscience and novel ideas are necessary [12]. Meanwhile, university students can be seen as indirect victims of the economy, as the global economic downturn has led to reduced employment opportunities for Korean university students, with an increase in the expenditures for new tools and books for online classes [34]. Korea’s economic recession caused by the COVID-19 pandemic has resulted in a high cost of living and increased tuition in colleges, which has become a high burden on low-income college students than before, bringing employment preparation and creating uncertainty about the future, which has negatively affected their mental health. Just as the Korean government has implemented consolation payments and proactive welfare policies for citizens, after the COVID-19 outbreak, the exploration and implementation of financial support policies, such as support for tuition and living expenses for university students are necessary.

This study found that the higher the academic stress, the lower the spiritual health, which was consistent with prior studies on the relationship between academic stress and spirituality in Korean nursing students [17,34]. The higher the life satisfaction and academic performance, the higher the spiritual health; this finding was similar to a previous study on nursing college students’ quality of life and spirituality [17] and University students’ academic performance and spirituality [35]. As there are complex factors that affect spiritual health, future studies could benefit from investigating the correlation between these variables.

Among the main variables, academic stress, life satisfaction, and academic performance were identified as factors influencing spiritual health. This was consistent with the results of studies that found the stress of university life and general stress of Korean undergraduate nursing students as factors negatively influencing spirituality before and after the COVID-19 outbreak [11,17]. In addition, this result was similar to the results of an international study, in which the academic burden of university students was shown to be a factor influencing mental health [28,36]. Particularly for nursing students, the stress regarding theoretical courses and clinical practice increased, which could have negatively impacted spiritual health before the COVID-19 outbreak [1,25]. However, as the spiritual health of undergraduate nursing students decreased during the pandemic [5,33], it would be advantageous for future measures to explore spiritual health by identifying characteristics of academic and general stress in nursing students, thereby enabling the formulation of coping strategies for during and post-pandemic periods.

Life satisfaction was identified as a factor positively influencing spiritual health, consistent with a previous study that found the daily life satisfaction of university healthcare students reduce the risk factors for mental health during the COVID-19 lockdown [37]. University students experience increased stress and decreased life satisfaction under the influence of COVID-19 [1,8,28,32]. To improve spiritual health among nursing students, it is essential that educational and counseling activities have greater scope for customizability. Moreover, the strengthening spirituality among students, as an important factor in health promotion, would lead to improvements in their mental health and happiness [38]. The level of the spiritual health of nursing students who have a high level of satisfaction in daily life with hope and a positive attitude despite the rapid changes caused by the external circumstances of COVID-19 was also high.

Furthermore, the below-3.0 grade point average (GPA) group was lower than that of students in the highest GPA group (4.0 and above), illustrating the negative influence low academic performance can have on the spirituality of undergraduate nursing students [11]. The level of spiritual health varies depending on the level of academic performance, which makes it necessary to develop programs according to academic achievement score. Among the characteristics of nursing students, Protestant and Catholic students had higher levels of spiritual health than Buddhist students or those without religion. This result was partially consistent with a previous study where the religion of university students in Canada had an impact on their spiritual quality of life after the COVID-19 outbreak [39]. This result implies that the influence of religion on spiritual health cannot be ruled out although “spiritual health” is a multidimensional and comprehensive concept that transcends specific religions. Furthermore, 70% of participants in this study were not religious and reported lower levels of spiritual health compared to those that were religious. Accordingly, it is necessary to consider students’ religion when developing a program to improve the spiritual health of nursing students in the future, such as programs for non-religious but spiritual individuals. 

In this study, fear of COVID-19 infection was not an influencing factor of spiritual health among the participants. This differed from the results of studies on anxiety and spirituality caused by the COVID-19 outbreak [8], as well as studies on Iranian undergraduate students [40]. This difference can potentially be explained according to the COVID-19 quarantine measures by country, the timing of the study, and the characteristics of the subject. Under the prolonged COVID-19 situation, the younger population is expected to feel more isolated and helpless [1,3]. Therefore, follow-up research is recommended to find ways to reduce academic stress and improve spiritual health by increasing life satisfaction through detailed evaluation and research on physical and mental health problems of nursing students who have experienced COVID-19. In addition, it is necessary to develop a curriculum that can accurately conceptualize and apply spiritual health in the education of nursing students, who will, in time, be working in the clinical field where the demand for the spiritual care of patients is increasing [12].

### Strengths and Limitations

The current study identifies the relevant factors related to the spiritual health of nursing students, providing basic data for education and counseling, two disciplines that are necessary for enabling an integrative state of health. However, this study has some limitations. First, the cross-sectional study design did not allow for causal inference. Second, convenience sampling was used, and the students were from the same colleges in Daegu; therefore, the sample was not fully representative of all regions of South Korea. Thus, large-scale studies with a longitudinal design and randomized sampling methods should be conducted in the future. Third, the spiritual health problems were self-rated by students, which may involve a recall bias. Further comparative studies are needed to better understand the underlying mechanisms of spiritual health problems in nursing students and their long-term impacts on the career development of the students, so as to provide policies or spiritual interventions in South Korea. 

## 5. Conclusions

Infectious diseases are expected to continue to occur globally, and the resulting development of medical technology demands that nurses in the clinical field should possess a higher level of spiritual care. Consequently, it is important that nurses should be provided with education and training that will help them to manage their spiritual health and maintain a sense of health and well-being through this, which requires continuous education and training from nursing students. Therefore, an appropriate intervention program such as meditation, prayer, and relaxation therapy to manage academic stress, academic performance, and satisfaction in life for enhancing spiritual health should be inculcated in the nursing curriculum to equip nursing students with internal resources to be successful holistic nursing care providers in the future.

## Figures and Tables

**Table 1 ijerph-20-03716-t001:** Spiritual health according to participants’ general characteristics (*N* = 219).

Variables	Categories	N (%)	Mean ± SD	t(f)	*p*	Scheffé
Sex	Male	33 (15.1)	3.64 ± 0.62	0.15	0.884	
	Female	186 (84.9)	3.62 ± 0.45			
Age (years)	19 ^a^	38 (17.4)	3.55 ± 0.40	3.64	0.007	a, b, c, d < e *
	20 ^b^	50 (22.8)	3.56 ± 0.58			
	21 ^c^	53 (24.2)	3.64 ± 0.35			
	22 ^d^	42 (19.2)	3.52 ± 0.47			
	≥23 ^e^	36 (16.4)	3.88 ± 0.51			
Religion	Protestantism	26 (11.9)	3.91 ± 0.52	4.45	0.005	
	Catholic	15 (6.8)	3.75 ± 0.56			
	Buddhism	24 (11.0)	3.64 ± 0.47			
	None	154 (70.3)	3.56 ± 0.45			
Socioeconomic status	High ^a^	13 (5.9)	3.85 ± 0.60	5.93	<0.001	a > d **
	High and Middle ^b^	64 (29.2)	3.77 ± 0.43			
	Middle ^c^	101 (46.1)	3.61 ± 0.40			
	Law ^d^	41 (18.7)	3.36 ± 0.58			
Monthly allowance (KRW 10,000)	<10	14 (6.4)	3.51 ± 0.65	2.20	0.070	
	10–19	36 (16.4)	3.43 ± 0.42			
	20–29	51 (23.3)	3.66 ± 0.34			
	≥30	109 (49.8)	3.69 ± 0.52			
Residential environment	Living with parents	131 (59.8)	3.62 ± 0.47	0.19	0.824	
	Lodging and dormitory	51 (23.3)	3.66 ± 0.43			
	Living alone	37 (16.9)	3.60 ± 0.60			
	Other	9 (4.1)	3.65 ± 0.54			
Grade	1	39 (17.8)	3.56 ± 0.42	1.24	0.295	
	2	49 (22.4)	3.61 ± 0.54			
	3	48 (21.9)	3.56 ± 0.46			
	4	83 (37.9)	3.70 ± 0.48			
Reason for entering nursing school	Entrance exam score	47 (21.5)	3.53 ± 0.55	1.54	0.192	
	Employment guarantee	77 (35.2)	3.59 ± 0.45			
	Aptitude	70 (32.0)	3.73 ± 0.45			
	Others’ recommendation	19 (8.7)	3.64 ± 0.57			
	Others	6 (2.7)	3.54 ± 0.23			
Leave of absence	Yes	25 (11.4)	3.84 ± 0.42	2.36	0.019	
	No	194 (88.6)	3.60 ± 0.48			
Club activities	Yes	134 (61.2)	3.68 ± 0.45	2.22	0.028	
	N0	85 (38.8)	3.53 ± 0.52			
Clinical practice	Yes	150 (68.5)	3.64 ± 0.51	0.83	0.408	
	No	69 (31.5)	3.59 ± 0.42			
Types of lectures	Face-to-Face (offline)	63 (28.8)	3.55 ± 0.47	0.99	0.400	
	Remote (online)	52 (23.8)	3.66 ± 0.49			
	Hybrid (offline + online)	104 (47.5)	3.65 ± 0.47			
Number of consultations (interviews with professors (per semester)	0	14 (6.4)	3.46 ± 0.36	1.79	0.132	
	1	118 (53.9)	3.59 ± 0.52			
	2	62 (28.3)	3.73 ± 0.44			
	3	12 (5.5)	3.75 ± 0.39			
	>4	13 (5.9)	3.†48 ± 0.47			
Major satisfaction	Bad ^a^	7 (3.2)	2.98 ± 0.71	10.49	<0.001	a < b, c, d ***
	Middle ^b^	56 (25.6)	3.46 ± 0.49			
	Good ^c^	110 (50.2)	3.67 ± 0.43			
	Very Good ^d^	46 (21.0)	3.82 ± 0.42			
Academic performance (GPA)	<3.0 ^a^	11 (5.0)	3.26 ± 0.48	3.37	0.019	a < b<c, d †
	3.0 ≤ – <3.5 ^b^	59 (26.9)	3.56 ± 0.48			
	3.5≤ – < 4.0 ^c^	89 (40.6)	3.65 ± 0.44			
	≥4.0 ^d^	60 (27.4)	3.71 ± 0.51			

SD: Standard Deviation. *p*< 0.05. * As a Post hoc after ANOVA test (Scheffé) represents the significant difference in Spiritual Health (SH) by each age group, a, b, c, d and e mean the SH average of each age, indicating that the spiritual health level was lower than over the age of 23. ** a group (high socioeconomic status)’s SH level is significantly higher than d group(low) and no significant difference with b, c group. *** a group (‘bad’ of major satisfaction) is significantly lower than other groups (middle, good and very good). † c (GPA 3.5 ≤ – < 4.0) and d (GPA ≥ 4.0) groups’ SH level is significantly higher than a and b group.

**Table 2 ijerph-20-03716-t002:** Levels of main variables (*N* = 219).

Variables	Mean ± SD	Range	Scale Standardization	Range
Fear of COVID-19	19.31 ± 4.75	7–35	2.75 ± 0.68	1–5
Academic stress	56.28 ± 9.21	15–75	3.76 ± 0.61	1–5
Life satisfaction	19.96 ± 5.69	5–35	3.99 ± 1.14	1–7
Spiritual health	96.98 ± 11.54	24–120	3.63 ± 0.48	1–5

**Table 3 ijerph-20-03716-t003:** Correlations of main variables (*N* = 219).

	Spiritual Health	Fear of COVID-19	Academic Stress	Life Satisfaction	Academic Performance
	r(p)	r(p)	r(p)	r(p)	r(p)
Spiritual health	1				
Fear of COVID-19	−0.04	1			
Academic stress	−0.31 *	0.27 *	1		
Life satisfaction	0.43 *	−0.02	−0.3 *	1	
Academic performance	0.19 *	−0.08	−0.21 *	0.18 *	1

* *p* < 0.001.

**Table 4 ijerph-20-03716-t004:** Factors affecting spiritual health among nursing students (*N* = 219).

	Model 1				Model 2			
	B	SE	*β*	t(*p*)	B	SE	*β*	t(*p*)
(Constant)	2.59	0.42		6.23 (0.000)	2.67	0.49		5.39 (<0.001)
Age	0.02	0.02	0.09	1.27 (0.207)	0.02	0.02	0.09	1.36 (0.175)
D. Protestant	0.26	0.09	0.18	2.75 (0.006)	0.27	0.09	0.18	2.96 (0.003)
D. Catholic	0.23	0.12	0.12	1.99 (0.048)	0.28	0.11	0.15	2.45 (0.015)
D. Buddhist	0.07	0.09	0.04	0.66 (0.510)	0.08	0.09	0.05	0.87 (0.387)
D. Club activity	0.09	0.06	0.09	1.46 (0.146)	0.07	0.06	0.07	1.084 (0.280)
D. Socioeconomic status_ High and Middle	−0.02	0.12	−0.02	−0.19 (0.854)	−0.02	0.12	−0.02	−0.13 (0.893)
D. Socioeconomic status_ Middle	−0.11	0.12	−0.11	−0.90 (0.371)	−0.09	0.11	−0.09	−0.85 (0.397)
D. Socioeconomic status_ Low	−0.33	0.13	−0.26	−2.53 (0.012)	−0.22	0.13	−0.17	−1.71 (0.088)
D. Leave of absence	0.13	0.10	0.09	1.29 (0.199)	0.14	0.09	0.09	1.43 (0.153)
Major Satisfaction	0.19	0.04	0.29	4.72 (<0.001)	0.08	0.04	0.13	1.96 (0.052)
Fear of COVID-19					0.02	0.04	0.03	0.45 (0.656)
Academic Stress					−0.26	0.15	−0.15	−2.21 (0.045)
Life Satisfaction					0.11	0.03	0.27	3.85 (<0.001)
D. GPA < 3.0					−0.30	0.14	−0.14	−2.08 (0.039)
D. GPA 3.0 ≤ – <3.5					−0.01	0.08	−0.01	−0.09 (0.929)
D. GPA 3.5 ≤ – <4.0					0.01	0.07	0.01	0.14 (0.891)
	R = 0.251, Adj R = 0.226 F(*p*) = 7.94 (<0.001)				R = 0.35.1, Adj R = 0.307 F(*p*) = 6.83 (<0.001)			

D: Dummy variables reference group: religion (none), socioeconomic status (high), leave absence (yes), academic performance (GPA ≥ 4.0). Note. The unstandardized beta (B), the standard error for the unstandardized beta (SE), the standardized beta (β), the *t*-test statistic (t), and the probability value (*p*). *p* < 0.05.

## Data Availability

Not applicable.

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
