# Peer review of "Factors Influencing Spiritual Health among Nursing Students in the Prolonged COVID-19 Situation"

_ijerph, 2023, doi:10.3390/ijerph20043716_

Round 1

Reviewer 1 Report

1) Introduction: As rightfully pointed out by the author, the term "spiritual health" is a very arbitrary concept. Although there's an attempt to explain this in the manuscript, I think its still insufficient to address confusion. The definition used by the author as defined by Choi and Kim (2021) is poorly defined when the term "spiritual" is used twice in the definition without actually explaining what "spiritual" means. Also, there needs to be justification why this paper's definition was selected. Since the crux of this paper lies in this definition, more attention needs to be placed in this regard.

2) Instruments: There needs to be justification on why these instruments were selected and its purpose in the study.

3) Spiritual Health Scale-short form (SHS-SF): There needs to be more explanation on this since this is how the main study objectives is achieved - i.e. what does this instrument actually evaluate (its components), and how are the scores interpreted.

4) Discussion: The discussion seems to be on the surface level, whereby it only covers the correlation between different factors with perceived spiritual health. And as expected poor socio-economic circumstances, high academic stress, low life satisfaction all had a negative impact on spiritual health - but how did each of this specifically effect the spirituality of the participants? I think the confusion lies due this goes back to "spiritual health" being poorly defined in the first place, because currently it appears to be akin to general "mental health".

5) Conclusion: Is there validation from this study that "spiritual resources, such as meditation, prayer, and relaxation therapy" improves spiritual health?

Author Response

Thank you for giving me the opportunity to submit a revised draft of my manuscript titled Factors Influencing Spiritual Health Among Nursing Students in the Prolonged COVID-19 Situation to International Journal of Environmental Research and Public Health. I appreciate the time and effort that you and the reviewers have dedicated to providing your valuable feedback on my manuscript. I am are grateful to the reviewers for their insightful Comments on my paper. I have been able to incorporate changes to reflect most of the suggestions provided by the reviewers. I have highlighted the changes within the manuscript. 

Reviewer1

1) Introduction:

As rightfully pointed out by the author, the term "spiritual health" is a very arbitrary concept. Although there's an attempt to explain this in the manuscript, I think its still insufficient to address confusion. The definition used by the author as defined by Choi and Kim (2021) is poorly defined when the term "spiritual" is used twice in the definition without actually explaining what "spiritual" means. Also, there needs to be justification why this paper's definition was selected. Since the crux of this paper lies in this definition, more attention needs to be placed in this regard.

I thank the reviewer for this pertinent comment I have revised previous definition of “spiritual health” and added more text for a clearer understanding of the concept in the introduction. See page 1-3, Introduction

2) Instruments: There needs to be justification on why these instruments were selected and its purpose in the study:

I added the following sentence in 2.3. Instruments

; The questionnaires used in this study elicited information on fear of COVID-19, academic stress, academic performance, satisfaction of life, spiritual health, and sociodemographic information. All questionnaires were developed in Korean and were completed by Korean-speaking participants. Page 3. Lines 130-134

And also added a sentence each instrument as follows, this is a sample. Page 3, under line135 :  

 2.3.2. Scale of Academic Stress (SAS)

             Academic stress was measured with SAS developed for adolescents by Park and Park (2012) [21], and included “parents,” “teacher,” and “oneself” as the causes of academic stress. However, in this study, based on previous studies regarding nursing students, 15 questions were measured using only “oneself” as a source of academic stress. The scale contained 15 items on a five-point Likert scale and the score ranged from 15–75 points with higher scores indicating a higher level of academic stress. Cronbach’s α was 0.85 in Lee and Park’s study for nursing students [18] and 0.87 in this study.

3) Spiritual Health Scale-short form (SHS-SF): There needs to be more explanation on this since this is how the main study objectives is achieved - i.e. what does this instrument actually evaluate (its components), and how are the scores interpreted.

I quite agree with you. I have added more text to explain the Spiritual Health Scale-short form (SHS-SF) as follows: “The score ranged from 24–120 points; with higher scores indicating a higher level of spiritual health.” Page 4, lines 165-177

Participants’ religious affiliation in the demographic characteristics is not a measure of spirituality. Therefore, this study adapted SHS-SF developed by Hsiao et al. (2013) which had high validity and reliability for measuring the spiritual health of students and registered nurses in Taiwan which has a Buddhist and Confucian culture like that of Korea [25]. There are several instruments of spirituality in previous studies including the Spiritual Well Being Scale (SWB), the JAREL spiritual well-being scale, the Functional Assessment of Chronic Illness Therapy Spiritual Well-being scale (FACIT-Sp), and Spiritual Index of Well-being (SIWB). These instruments generally have common shortcomings. The majority of scale items are focused on religiosity, which may not be applicable to non-religiosity and all instruments were developed based on western Judeo-Christian society [25]. This study utilized the SHS-SF which can measure the concept of spiritual health distinct from spirituality and spiritual well-being; moreover, it was first translated into Korean and applied

4) Discussion: The discussion seems to be on the surface level, whereby it only covers the correlation between different factors with perceived spiritual health. And as expected poor socioeconomic circumstances, high academic stress, low life satisfaction all had a negative impact on spiritual health - but how did each of this specifically effect the spirituality of the participants? I think the confusion lies due this goes back to "spiritual health" being poorly defined in the first place, because currently it appears to be akin to general "mental health" :

Thank you for the comment. To address your comment, I added the following text in the Discussion section for rigor.  

- pp.10, lines 292-296: In particular, students with no religious affiliation had the lowest level of spiritual health. This means that not only religiosity but also non-religiosity is an important factor in spiritual health; therefore, education and training about the spiritual realm, beliefs, values, and ethics derived from human conscience and novel ideas are necessary [12].

-pp.10, lines 299-302: Korea's economic recession caused by the COVID-19 pandemic has resulted in a high cost of living and increased tuition in colleges, which has become a high burden on low-income college students than before, bringing employment preparation and creating uncertainty about the future, which has negatively affected their mental health

-pp.11, lines 334-338: Moreover, the strengthening spirituality among students, as an important factor in health promotion, would lead to improvements in their mental health and happiness [38]. The level of the spiritual health of nursing students who have a high level of satisfaction in daily life with hope and a positive attitude despite the rapid changes caused by the external circumstances of COVID-19 was also high.

5) Conclusion: Is there validation from this study that "spiritual resources, such as meditation, prayer, and relaxation therapy" improves spiritual health?

In line with your question, I have modified the Conclusion section. Based on the reviewer's comments and the purpose and results of this study, I amended the conclusion as follows, page 12, line 382-392;

Conclusions (original version)

Nurses who engage in the direct treatment and care for patients with infectious diseases during pandemics should be able to increase their life satisfaction using spiritual resources, such as meditation, prayer, and relaxation therapy, to maintain and promote spiritual health. In doing so, the quality of life, as well as quality of care, can be improved. Therefore, education and training for reducing academic stress and in-creasing life satisfaction to manage one's spiritual health are necessary.

Conclusions (revision version)

Infectious diseases are expected to continue to occur globally, and the resulting development of medical technology demands that nurses in the clinical field should possess a higher level of spiritual care. Consequently, it is important that nurses should be provided with education and training that will help them to manage their spiritual health and maintain a sense of health and well-being their spiritual health and maintain a sense of overall health and well-being through this, which requires continuous education and training from nursing students. Therefore, an appropriate intervention program such as meditation, prayer, and relaxation therapy to manage academic stress, academic performance, and satisfaction in life for enhancing spiritual health should be inculcated in the nursing curriculum to equip nursing students with internal resources to be holistic nursing care providers in the future.

Reviewer 2 Report

I suggest making the following fixes:

In the Materials and Methods section:

1. Connection 2.1. Study design and 2.2. samples

2. Description of how the (random?) sample was selected

The largest group of None should be included in the analyzes of the religion variable.

I suggest adding information in the text that model 1 includes socio-demographic variables and model 2 includes variables related to spiritual health. Are all variables (e.g. Club activities) necessary?

The mediating role of variables can be used to explain the obtained results.

Regards

Author Response

Dear reviewers,

Thank you for giving me the opportunity to submit a revised draft of my manuscript titled Factors Influencing Spiritual Health Among Nursing Students in the Prolonged COVID-19 Situation to International Journal of Environmental Research and Public Health. I appreciate the time and effort that you and the reviewers have dedicated to providing your valuable feedback on my manuscript. I am grateful to the reviewers for their insightful Comments on my paper. I have been able to incorporate changes to reflect most of the suggestions provided by the reviewers. I have highlighted the changes within the manuscript. 

Reviewer 2

In the Materials and Methods section:

  1. Connection 2.1. Study design and 2.2. samples

: Thank you for your observation. I have revised this in line with your comment. See the page 3, line 107

  1. Description of how the (random?) sample was selected

: The following sentences were added and modified according to the reviewer's comment.  pp.3. line 116

2.2 Data collection: A convenience sampling was applied to select nursing students from three consenting nursing colleges in Daegu Metropolitan City from September 2 to 18.

  1. The largest group of None should be included in the analyzes of the religion variable.

 : Thank you for your comment. I have added it in the Discussion as follows, page 10, lines 292-295

 In particular, students with no religious affiliation had the lowest level of spiritual health. This means that not only religiosity but also non-religiosity is an important factor in spiritual health; therefore, education and training on the spiritual realm belonging to the realm of beliefs, values, and ethics derived from human conscience and novel ideas are necessary [12].

  1. I suggest adding information in the text that model 1 includes socio-demographic variables and model 2 includes variables related to spiritual health. Are all variables (e.g. Club activities) necessary?

: Thank you for your comment. In this study, the variables input to the first step of the hierarchical regression analysis were variables that showed significant differences in spiritual health in Table 1, and were used as control variables to confirm the influence of the main variables, page 8, lines 236-243

…In step1, participants’ age, religion, socioeconomic status, leave of absence, experience in club activities, and major satisfaction among demographic factors, which statistically significantly differed according to spiritual health were entered into Model 1 for hierarchical regression and nominal variables were dummy coded. The explanatory power of Model 1 was 22.6% and the model had a significant fit (F= 7.94, p < 0.001). In step 2, fear of COVID-19, academic stress, academic performance levels, and life satisfaction were additionally entered as independent variables after controlling for general characteristics in Model 2. The spiritual health of students with being Protestant (β =0.18) and Catholic (β =0.15) was higher than in the non-religious group (referenced value) and that of individuals with grades below 3.0 (β =-0.14) was lower than for those with grades over 4.0 (referenced value). Academic stress(β=-0.15) and performance (<3.0 grades) were negative predictors, and satisfaction of life(β=0.27) and being religious (Protestant and Catholic) were positive predictors of spiritual health. The explanatory power of Model 2 was 30.7% and the model had a significant fit (F = 6.83, p < 0.001).

  1. The mediating role of variables can be used to explain the obtained results

Round 2

Reviewer 2 Report

The changes made have improved. Congratulations.